# In Vitro Evaluation of Antimicrobial Amyloidogenic Peptides for the Treatment of Early and Mature Bacterial Biofilms

**DOI:** 10.3390/ijms26188767

**Published:** 2025-09-09

**Authors:** Pavel A. Domnin, Sergei Y. Grishin, Alexey K. Surin, Svetlana A. Ermolaeva, Oxana V. Galzitskaya

**Affiliations:** 1Gamaleya Research Centre of Epidemiology and Microbiology, 123098 Moscow, Russia; paveldomnin6@gmail.com; 2Biology Faculty, Lomonosov Moscow State University, 119991 Moscow, Russia; 3Institute of Protein Research, Russian Academy of Sciences, 142290 Pushchino, Russia; syugrishin@gmail.com; 4The Branch of the Institute of Bioorganic Chemistry, Russian Academy of Sciences, 142290 Pushchino, Russia; alan@vega.protres.ru; 5State Research Center for Applied Microbiology and Biotechnology, 142279 Obolensk, Russia; 6Institute of Environmental and Agricultural Biology (X-BIO), Tyumen State University, 625003 Tyumen, Russia; 7Institute of Theoretical and Experimental Biophysics, Russian Academy of Sciences, 142290 Pushchino, Russia

**Keywords:** antimicrobial peptides, amyloidogenic regions, cell-penetrating peptide, TAT fragment, Antp fragment, ribosomal S1 protein, biofilm-associated pathogens

## Abstract

Biofilm formation by pathogenic bacteria, including methicillin-resistant *Staphylococcus aureus* (MRSA), *Pseudomonas aeruginosa*, and *Escherichia coli*, represents a major clinical challenge due to the high resistance of biofilms to conventional antimicrobial therapy. In this in vitro study, we investigated the antimicrobial and antibiofilm activity of synthetic peptides R23I^T^, R23L^P^, V31K^T^, R44K^S^, R44K^P^, V31K^S^, and I31K^P^ against methicillin-resistant *S. aureus* (MRSA, SA180-F strain), *S. aureus* (129B), *P. aeruginosa* (2943), and *E. coli* (MG1655). In liquid medium, peptides R23L^P^ and R44K^S^ exhibited the broadest and most potent antimicrobial activity against all tested strains. On solid agar, these peptides demonstrated comparable activity, with notable effects particularly against *E. coli*. We further assessed the peptides’ impact on both early-stage and mature biofilms using crystal violet staining for total biomass and the MTT assay for cellular metabolic activity. Peptide R44K^S^ showed a strong dose-dependent inhibitory effect on early MRSA biofilm formation, while most peptides unexpectedly enhanced biofilm formation by *S. aureus* and *E. coli*. Peptides R44K^P^ and V31K^S^ at 10 mg/mL significantly reduced both biomass and metabolic activity of early *P. aeruginosa* biofilms. None of the peptides inhibited mature biofilm biomass across species; however, several, particularly I31K^P^, significantly reduced the metabolic activity of MRSA within mature biofilms. These findings underscore the strain- and stage-specific effects of antimicrobial peptides and highlight R23L^P^, R44K^S^, R44K^P^, V31K^S^, and I31K^P^ as promising candidates for targeted biofilm control in vitro, especially against MRSA.

## 1. Introduction

The rapid global rise of microbial resistance to conventional antibiotics poses a critical threat to public health, limiting the effectiveness of existing therapies and leading to persistent infections, prolonged hospital stays, and increased mortality [1,2,3]. This is particularly evident in biofilm-associated infections and in cases involving multidrug-resistant pathogens such as methicillin-resistant *S. aureus* (MRSA) and *P. aeruginosa* [4,5,6]. In response to the urgent need for novel antimicrobial strategies, antimicrobial peptides (AMPs) have emerged as promising therapeutic agents due to their broad-spectrum activity, rapid bactericidal effects, and reduced susceptibility to resistance development. Naturally occurring or synthetically engineered AMPs can disrupt bacterial membranes, interfere with intracellular targets, and, in some cases, modulate host immune responses. Their unique mechanisms of action offer a valuable alternative to traditional antibiotics, especially in the context of combating drug-resistant infections and disrupting microbial biofilms [7,8].

Bacterial biofilms are structured communities of microorganisms embedded in a self-produced extracellular matrix that provides protection from environmental stressors, including antimicrobial agents and host immune responses [9,10,11]. Biofilm-associated infections represent a significant clinical challenge, particularly in hospital settings, where they contribute to chronic wounds, catheter-associated infections, and implant-related complications [12,13]. Among the most problematic pathogens are methicillin-resistant *S. aureus* (MRSA) and multidrug-resistant Gram-negative species such as *P. aeruginosa* and *E. coli*, which are frequently implicated in recalcitrant biofilm-associated infections [14,15].

Despite ongoing advances in antimicrobial drug development, traditional antibiotics are often ineffective in eradicating biofilms [16,17]. This inefficacy is largely due to the inherent differences between planktonic and sessile bacterial states [18]. In biofilms, bacteria exhibit reduced metabolic activity, upregulate stress response pathways, and exist within a dense polymeric matrix that impedes antibiotic diffusion.

Given the intrinsic resilience of biofilms and the limited efficacy of current antimicrobial agents in penetrating or disrupting their complex architecture, there is an urgent need for innovative therapeutic approaches specifically designed to target the biofilm mode of microbial growth. In this regard, antimicrobial peptides (AMPs) have emerged as increasingly promising candidates for anti-biofilm therapy. Both naturally occurring and synthetically engineered AMPs have demonstrated the capacity to inhibit initial microbial adhesion, prevent the maturation of biofilms, and eradicate bacteria embedded within the biofilm matrix [19]. It has previously been demonstrated that biofilms represent a primary target that is highly resistant to conventional disinfectants and antibiotics. In this context, antimicrobial (anti-biofilm) peptides are considered particularly promising due to their activity against a broad spectrum of pathogens, including Gram-positive and Gram-negative bacteria as well as fungi [20].

Bacterial biofilms, particularly those formed by pathogens such as MRSA and *P. aeruginosa*, are characterized by high resistance to both antibiotics and the host immune response. It is well established that the structural stability of bacterial biofilms is primarily maintained through the production of extracellular polymeric substances, modifications of the cell wall, and the upregulated expression of specific proteins [21,22]. It is believed that functional amyloid proteins produced by bacteria are integral components of the extracellular polymeric matrix of biofilms, providing them with structural stability [23,24,25,26,27]. It is hypothesized that the introduction of exogenous amyloidogenic peptides may interfere with these processes [28]. In particular, peptide co-aggregation with the S1 ribosomal protein may reduce the viability of bacterial cells within the biofilm, thereby weakening its integrity. Although direct experimental evidence on the disruption of biofilms by S1-targeting peptides is currently lacking, conceptually similar antimicrobial peptides—such as those derived from Aβ sequences or dermaseptins—have demonstrated both antimicrobial and anti-biofilm activity [29]. Therefore, it is reasonable to expect that such peptides may at least inhibit bacterial growth within the biofilm or facilitate the penetration of other antibacterial agents. Furthermore, the combination of an amyloid-based mechanism with membrane-disruptive action may enhance the ability to break down microcolonies within the biofilm structure. In addition to direct membrane disruption, amyloid peptides exhibit antimicrobial activity through several alternative mechanisms. These include the induction of intracellular protein aggregation within bacterial cells, ultimately leading to cell death; the binding of amyloid structures to microbial proteins, resulting in conformational alterations that inhibit their function; and the agglutination of microbial cells into large, non-functional aggregates that impair their viability [30].

The development of complex microbial biofilms can be broadly divided into four distinct stages: (1) the initial attachment of planktonic cells to a surface, (2) cell aggregation and the formation of microcolony-like structures (early biofilm development), (3) the maturation of species-specific biofilm architecture, and (4) the dispersal of biofilm-resident cells back into the planktonic phase [31,32,33]. Accordingly, the evaluation of the antibiofilm potential of an antimicrobial peptide should not be limited to its ability to inhibit biofilm formation during the early stages, but must also include assessment of its efficacy against mature, established biofilms during the later stages of development.

The aim of this study was to investigate the antimicrobial activity of peptides R23I^T^, R23L^P^, V31K^T^, R44K^S^, R44K^P^, V31K^S^, and I31K^P^ against biofilms formed by Gram-positive *S. aureus* (methicillin-sensitive (MSSA) strain 129B and methicillin-resistant (MRSA) strain SA180-F), as well as by Gram-negative *P. aeruginosa* (strain 2943) and *E. coli* (strain MG1655). The designed peptides were derived from amyloidogenic sequences identified within the S1 ribosomal protein fragments of bacterial species [34,35]. Each peptide includes one or two cell-penetrating peptide (CPP) motifs—either the TAT peptide (RKKRRQRRR) [36] or the antennapedia peptide (RQIKIWFQNRRMKWKK) [37]—linked to the amyloidogenic fragment via glycine-rich linkers, specifically R23I^T^ [38] and V31K^T^ from *T. thermophilus*; R23L^P^, R44K^P^, and I31K^P^ from *P. aeruginosa* [39]; and R44K^S^ and V31K^S^ from *S. aureus* [40]. Special attention was given to assessing peptide efficacy at different stages of biofilm development, including both early-stage (initial attachment and microcolony formation) and late-stage (mature) biofilms.

## 2. Results

### 2.1. Determination of Antibacterial Properties of Peptides in Liquid Medium

We evaluated the antibacterial activity of the tested peptides R23I^T^, R23L^P^, V31K^T^, R44K^S^, R44K^P^, V31K^S^, and I31K^P^ in liquid medium against MRSA (SA 180-F strain), *S. aureus* (129B strain), *E. coli* (MG1655 strain), and *P. aeruginosa* (2943 strain) (Table 1, Appendix A).

According to the data presented in Table 1, peptides R23L^P^ and R44K^S^, exhibited a broad spectrum of activity in liquid medium. These peptides demonstrated antimicrobial activity against all tested bacterial strains at concentrations ranging from 0.1 to 10 mg/mL. At the same time, the antimicrobial activity of peptides R23L^P^ and R44K^S^ at a concentration of 10 mg/mL against *E. coli* was comparable to that of meropenem. The least active peptide was V31K^T^.

### 2.2. Results of Antimicrobial Activity Testing of Peptides on Agar

Drops of the samples were applied to the surface of solidified 0.7% agar at the concentrations indicated in Table 2 and Appendix A. After 24 h of incubation, the results were evaluated based on the presence or absence (+/−) of inhibition zones.

Thus, antibacterial activity assessed by the agar diffusion method was observed only at high concentrations (1 mg/mL) of the peptides R23I^T^, R23L^P^, R44K^S^, R44K^P^, and V31K^S^. When the peptide concentration was reduced to 0.1 mg/mL or lower, no antimicrobial effect was detected against MRSA, *S. aureus*, *E. coli*, and *P. aeruginosa*.

### 2.3. Effect of Peptides on Early Biofilm Formation

We systematically evaluated the effects of peptides R23I^T^, R23L^P^, V31K^T^, R44K^S^, R44K^P^, V31K^S^, and I31K^P^ on both early and late stages of biofilm formation in MRSA (SA 180-F strain), *S. aureus* (129B strain), *E. coli* (MG1655 strain), and *P. aeruginosa* (2943 strain) using crystal violet staining and the MTT assay.

The results of peptide testing R23L^P^, V31K^T^, R44K^S^, and R44K^P^ against early-stage MRSA biofilms, based on the optical density values obtained after crystal violet staining, are shown in Figure 1.

As shown in Figure 1, among the peptides R23L^P^, V31K^T^, R44K^S^, and R44K^P^, only the peptide R44K^S^ exhibited an inhibitory effect on MRSA biofilm formation compared to the negative control (bacterial culture), which was confirmed by a decrease in crystal violet staining intensity with increasing peptide concentration (1 and 10 mg/mL) (Figure 1C). Peptides R23L^P^ and V31K^T^ demonstrated an inhibitory effect at a low concentration of 0.01 mg/mL; however, with increasing concentration, the inhibitory effect was absent or reversed (as observed for V31K^T^ at 10 mg/mL) (Figure 1A,B). At the same time, all peptides R23L^P^, V31K^T^, R44K^S^, and R44K^P^ at the tested concentrations were less effective compared to meropenem.

Overall data on the activity of peptides R23I^T^, R23L^P^, V31K^T^, R44K^S^, R44K^P^, V31K^S^, and I31K^P^ against early-stage biofilm formation by MRSA and other bacterial cultures (*S. aureus*, *E. coli*, and *P. aeruginosa*), assessed using crystal violet staining and the MTT assay, are presented in Table 3 and Appendix A.

As shown in Table 3, during the early stages of biofilm formation by *S. aureus*, all tested peptides failed to exhibit inhibitory activity based on crystal violet staining. On the contrary, an enhancement of biofilm formation was observed in the presence of peptides. According to the MTT assay, the peptides supported the maintenance of *S. aureus* metabolic activity.

For MRSA, crystal violet staining indicated a marked reduction in biofilm biomass following treatment with R23L^P^, R44K^S^, R44K^P^, and V31K^S^. The MTT assay revealed not only preservation of bacterial metabolic activity after peptide exposure but also a tendency toward increased metabolic activity in MRSA cells.

In the case of *E. coli*, crystal violet staining showed no inhibitory effect on biofilm formation. On the contrary, the optical density of peptide-treated samples increased compared to the negative control. Moreover, there was a concentration-dependent increase in biofilm biomass (Appendix A). Similarly, MTT assay data indicated that the peptides did not suppress the viability of *E. coli* cells. Thus, the peptides did not exhibit inhibitory activity against the early stages of *E. coli* biofilm development.

According to crystal violet data, peptides R23L^P^, V31K^T^, R44K^S^, R44K^P^, and V31K^S^ exerted inhibitory effects on early-stage biofilm formation by *P. aeruginosa* strain 2943. Notably, peptides R44K^P^ and V31K^S^ at 10 mg/mL not only suppressed early biofilm biomass accumulation but also reduced bacterial metabolic activity, as confirmed by the MTT assay results.

### 2.4. Effect of Peptides on Mature Biofilm Formation

The effects of peptides on the late stages of biofilm formation are presented in Table 4. Peptide preparations and meropenem at concentrations of 0.01, 0.1, 1, and 10 mg/mL were added after 24 h of incubation, that is, to already-formed mature bacterial biofilms.

According to the results of peptide testing and crystal violet staining, all tested peptides at concentrations of 0.01, 0.1, 1, and 10 mg/mL did not exhibit any inhibitory activity against mature biofilms.

At the same time, MTT assay data, which reflect the metabolic activity of bacteria within mature biofilms, revealed differences in peptide effects depending on the bacterial species. Peptides R23I^T^, R23L^P^, V31K^T^, R44K^S^, R44K^P^, and V31K^S^ reduced the metabolic activity of MRSA strain SA 180-F cells. Peptide I31K^P^ decreased the metabolic activity of both *S. aureus* and MRSA cells at concentrations of 0.01, 0.1, 1, and 10 mg/mL. However, the extent of metabolic inhibition caused by the peptides was lower than that observed for the antibiotic meropenem (Appendix A).

Meanwhile, peptides R23I^T^, R23L^P^, V31K^T^, R44K^S^, R44K^P^, and V31K^S^ did not inhibit the residual viability of *E. coli* and *P. aeruginosa* cells.

## 3. Discussion

Bacterial biofilms are highly organized communities of microorganisms embedded within an extracellular polysaccharide matrix. This structure provides bacteria with protection against antimicrobial agents and the host immune system, making biofilms a key pathogenic factor in chronic and nosocomial infections [41,42].

In the present study, a comprehensive evaluation was conducted to assess the antimicrobial and antibiofilm activities of synthetic peptides R23I^T^, R23L^P^, V31K^T^, R44K^S^, R44K^P^, V31K^S^, and I31K^P^ against both Gram-positive and Gram-negative pathogens. Special emphasis was placed on comparing their effects during the early and late stages of biofilm formation. The crystal violet staining method allows for the quantification of total biofilm biomass, including both live and dead cells, as well as the extracellular polymeric matrix. This assay reflects the structural presence of the biofilm but does not indicate its viability [43]. In contrast, the MTT assay assesses the metabolic activity of viable cells within the biofilm, as it relies on the reduction of tetrazolium salts by metabolically active cells [44]. Thus, crystal violet provides information on the physical presence of the biofilm, whereas MTT reflects the viability of the cells within it. The most reliable assessment is achieved by using both methods in combination, especially when assessing antimicrobial peptides, which may have different effects on the structure and viability of bacterial biofilms. This integrated approach helps avoid false-positive or false-negative interpretations, where one method may underestimate the peptide’s efficacy, while the other may overestimate it. For example, a decrease in optical density in the crystal violet assay does not necessarily indicate cell death if metabolic activity in the MTT assay remains high. It is also possible that the observed increase in biofilm biomass detected by crystal violet staining may be partially explained by the direct incorporation of peptides into the extracellular matrix. Such incorporation could enhance dye binding and retention, leading to an overestimation of actual biofilm growth. Conversely, cells may be non-viable while the matrix remains intact—this would be evident in the crystal violet assay, but not in the MTT assay. The use of two complementary methods, crystal violet staining and the MTT assay, enabled differentiation between the physical presence of biofilm biomass (including dead cells and matrix components) and the metabolic activity of viable cells within the biofilms. This approach is critically important for the accurate interpretation of peptide antimicrobial activity, as certain peptides may disrupt biofilm structure without affecting cell viability, and vice versa.

Among the tested peptides, R44K^S^ demonstrated the highest efficacy at the early stages of MRSA (SA 180-F strain) biofilm formation, causing a dose-dependent reduction in biofilm biomass. In contrast, all peptides tested against early-stage *S. aureus* (129B strain) biofilms exhibited a paradoxical effect: an enhancement of biofilm formation was observed. This effect could be partly explained by peptide incorporation into the matrix. Alternatively, more bacteria could enter the biofilm to avoid direct interactions with peptides in the liquid phase. Some quorum-sensing mimicry effects could be suggested in *S. aureus*, as Gram-positives use peptides as quorum sensing signals. Overall, the peptides were more effective against early-stage MRSA biofilms compared to their effects on *S. aureus*. Similar differences were noted between *P. aeruginosa* (generally more responsive to peptide treatment) and *E. coli* (largely unresponsive). These differences may be attributed, in part, to variations in biofilm porosity and density across bacterial species, which can influence peptide diffusion [45,46]. MRSA and *P. aeruginosa* may form less dense matrices at early stages compared to the mature biofilms of *S. aureus* and *E. coli*. In particular, it has been demonstrated that *E. coli* initiates biofilm formation more efficiently and transitions to the biofilm lifestyle significantly earlier than *S. aureus* and *P. aeruginosa* [47]. More broadly, differences in biofilm susceptibility to peptides likely result from a combination of factors, including cell wall structure, regulatory mechanisms, biofilm architecture, developmental stage, and stress response pathways [48].

Peptides did not exhibit significant disruptive effects on mature biofilms, as assessed by crystal violet staining. However, MTT assay results revealed a reduction in metabolic activity in MRSA cells following treatment with several peptides (R23L^P^, V31K^T^, R44K^S^, R44K^P^, V31K^S^, and I31K^P^), suggesting a partial bacteriostatic effect that was not accompanied by matrix degradation. Of particular interest is the opposite trend observed for *E. coli* (MG1655 strain) and *P. aeruginosa* (2943 strain), where peptide exposure was associated with maintenance or even enhancement of metabolic activity. This may reflect adaptive bacterial responses to stress or growth stimulation in response to sublethal peptide concentrations. At low or sublethal levels, antimicrobial peptides can induce protective mechanisms in bacteria, including increased matrix production and enhanced biofilm formation [49]. Furthermore, the tested peptides may be perceived by bacteria as signaling molecules that mimic quorum sensing effects and induce the expression of biofilm-associated genes [50].

It should be noted that the results of tests conducted on solid medium (agar) confirmed the antibacterial activity of peptides R23I^T^, R23L^P^, R44K^S^, R44K^P^, and V31K^S^ against MRSA and *S. aureus*, but only at a high concentration (1 mg/mL). When the dose was reduced to 0.1 mg/mL or lower, the antimicrobial effect was no longer observed. In liquid medium, the broadest spectrum of activity was demonstrated by peptides R23L^P^ and R44K^S^, which were effective against all tested bacterial strains. A comparison of the results obtained from biofilm assays, liquid medium testing, and agar diffusion indicates the need for either structural modification of the peptides to enhance their penetration capacity or the use of combined peptide treatments to achieve potential synergistic effects [51].

This study demonstrated that the antimicrobial peptides exhibited varying efficacy depending on the bacterial species and the stage of biofilm formation. The highest activity against early-stage biofilms of MRSA and *P. aeruginosa* was observed for peptides R23L^P^, R44K^S^, R44K^P^, and V31K^S^, whereas against *E. coli* and *S. aureus* most peptides were ineffective or even enhanced biofilm formation. None of the tested peptides induced substantial disruption of mature biofilms, as shown by crystal violet staining. However, a reduction in metabolic activity of MRSA cells, detected by the MTT assay, may indicate a bacteriostatic effect without matrix degradation.

These findings also highlight the importance of accounting for the specific structural and physiological features of bacterial biofilm matrices and stress responses when designing antibiofilm strategies. Given that most peptides exhibited significant activity only at high concentrations (1 mg/mL), future directions should focus on structural optimization—such as the incorporation of D-amino acids, cyclization, or N- and C-terminal modifications—to enhance peptide stability and penetration, as well as on exploring synergistic combinations with conventional antibiotics or other peptides with complementary mechanisms to improve biofilm disruption and antimicrobial efficacy. Overall, peptides R23L^P^, R44K^S^, R44K^P^, and V31K^S^ demonstrated the greatest potential as antimicrobial agents against MRSA biofilms, showing activity at early stages (based on crystal violet data) and at later stages (based on MTT assay results).

## 4. Materials and Methods

### 4.1. Peptide Synthesis and Preparation

Peptides R23I^T^ (RKKRRQRRRGGGGVTDFGVFVEI, Mr = 2675.1 Da) from S1 *T. thermophilus*, R23L^P^ (RKKRRQRRRGGGGITDFGIFIGL, Mr = 2645.1) from S1 *P. aeruginosa*, V31K^T^ (VTDFGVFVEIGGGGSRQIKIWFQNRRMKWKK, Mr = 3669.3 Da) from S1 *T. thermophilus*, R44K^S^ (RKKRRQRRRGGGGVVVHINGGKFGGGGSRQIKIWFQNRRMKWKK, Mr = 5163.1 Da) from S1 *S. aureus*, R44K^P^ (RKKRRQRRRGGGGITDFGIFIGLGGGGSRQIKIWFQNRRMKWKK, Mr = 5189.1 Da) from S1 *P. aeruginosa*, V31K^S^ (VVVHINGGKFGGGGSRQIKIWFQNRRMKWKK, Mr = 3613.3 Da) from S1 *S. aureus*, and I31K^P^ (ITDFGIFIGLGGGGSRQIKIWFQNRRMKWKK, Mr = 3639.3 Da) from S1 *P. aeruginosa* were obtained as commercial products (IQ Chemical LLC, S. Petersburg, Russia). According to the data provided by the supplier, the peptides were obtained with purity greater than 94.5%, as determined by HPLC. The identity of each peptide was confirmed by mass spectrometry (MALDI-TOF) performed by the supplier (Appendix A). Lyophilized peptides were delivered in powder form, stored at −20 °C.

### 4.2. Bacterial Strains

This study employed the following bacterial strains from the Gamaleya Center collection: a clinical isolate of methicillin-resistant *S. aureus* (MRSA), strain SA 180-F, resistant to benzylpenicillin and oxacillin (β-lactam antibiotics), ciprofloxacin, clindamycin, erythromycin, chloramphenicol, sulfamethoxazole, and vancomycin; a clinical isolate of *S. aureus* strain 129B, resistant to benzylpenicillin, clindamycin, erythromycin, oxacillin, sulfamethoxazole, and vancomycin; *P. aeruginosa* resistant strain 2943, resistant to gentamicin, ciprofloxacin, and trimethoprim; and *E. coli* susceptible strain MG1655. Strains were kept frozen in 10% glycerol at −80 °C until the experiments were started. After thawing, biochemical characteristics and the resistance profiles were confirmed.

### 4.3. Assessment of Antibacterial Activity of Peptides During Early Stages of Biofilm Formation

A stationary-phase culture of the tested bacterial strain was diluted 100-fold in fresh medium containing peptides or meropenem (positive control) at the specified concentrations in a volume of 200 µL per well in a 96-well round-bottom microtiter plate. BHI (Becton Dickinson, Franklin Lakes, NJ, USA) medium was used for *S. aureus* 129B and MRSA SA180-F strains and LB (Sigma-Aldrich, St. Louis, MO, USA) medium was used for *P. aeruginosa* and *E. coli* strain. After 24 h of incubation, the medium was removed. To assess the total biofilm biomass, the wells were rinsed with sterile PBS solution and stained with 0.1% crystal violet. The dye bound to the surface was then removed using 95% ethanol. To assess the presence of metabolically active cells, 0.3% MTT (PanEco Ltd., Moscow, Russia) solution was added to each well, and the plates were incubated at 37 °C for 2 h. After incubation, the MTT solution was removed, 150 µL of DMSO (Chimmed, Moscow, Russia) and 25 µL of glycine buffer were added to each well, and the plates were left at room temperature for 15 min. The resulting solutions were transferred to a flat-bottom 96-well plate, and optical density was measured at 595 nm.

### 4.4. Assessment of Antibacterial Activity of Peptides in Mature Biofilms

A stationary-phase culture of the tested strain was diluted 100-fold in fresh medium (200 µL per well) and added to the wells of a 96-well round-bottom plate. After 24 h of incubation, the medium was replaced with fresh medium containing the peptide or meropenem (positive control). After another 24 h, the medium was removed and the wells were rinsed with sterile PBS to eliminate planktonic cells. The procedures for assessing total biofilm biomass and cell metabolic activity using crystal violet and MTT assays were performed in the same manner as described above for early-stage biofilm experiments.

### 4.5. Determination of Antibacterial Activity of Peptides on 0.7% Agar

To assess the antibacterial activity of peptides using the agar diffusion method, a stationary-phase culture of the tested bacterial strains (MRSA strain SA180-F, *S. aureus* 129B, *P. aeruginosa* strain 2943, and *E. coli* strain MG1655) was diluted 10-fold in molten 0.7% agar and poured into sterile Petri dishes. Once the agar solidified, drops of peptide samples at the concentrations 1, 0.1, 0.01, and 0.001 mg/mL were applied to the agar surface.

### 4.6. Determination of Antibacterial Activity of Peptides in Liquid Medium

To evaluate the antibacterial activity of peptides in liquid culture, a stationary-phase culture of the tested bacterial strain was diluted 100-fold in fresh medium containing peptides or gentamicin (positive control) at the specified concentrations. A total volume of 200 µL was added to each well of a 96-well round-bottom microtiter plate. After 24 h of incubation at 37 °C, the samples were transferred to a flat-bottom 96-well microtiter plate, and the optical density was measured at 595 nm. Separately, the optical density of medium containing peptides but no bacteria was measured. These values were used as baseline (zero) readings for subsequent data analysis.

## 5. Conclusions

In conclusion, the obtained data demonstrate species-specific and stage-dependent efficacy of antimicrobial peptides against bacterial biofilms. The observed opposite effects on MRSA and *S. aureus*, *E. coli*, and *P. aeruginosa* suggest that peptides may induce adaptive responses or even stimulate biofilm formation depending on the concentration. These findings highlight the importance of rational peptide dosage and structural design, as well as the need for further studies on their mechanisms of action and potential synergy in combination therapies.

## Figures and Tables

**Figure 1 ijms-26-08767-f001:**
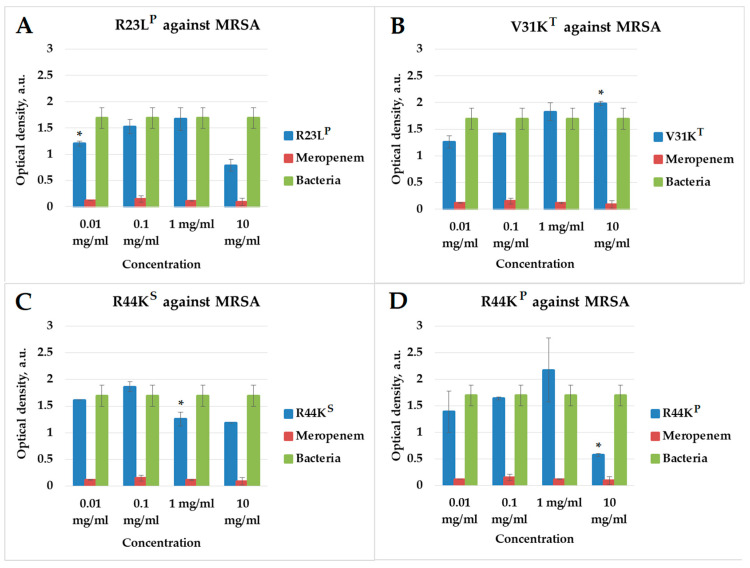
Antimicrobial effects of peptides R23L^P^ (**A**), V31K^T^ (**B**), R44K^S^ (**C**), and R44K^P^ (**D**) at concentrations of 0.01, 0.1, 1, and 10 mg/mL against early-stage biofilm formation of MRSA (SA 180-F strain). The antibiotic meropenem was used as a positive control for antimicrobial activity. Bacterial cultures grown without the addition of antimicrobial agents served as the negative control. All experiments were conducted in two independent replicates, and the data are presented as mean values with standard deviation (mean ± SD). *—indicates a statistically significant difference from the negative control (*p* < 0.05).

**Table 1 ijms-26-08767-t001:** Results of peptide testing in liquid medium.

Peptide	MRSA (SA 180-F Strain)	*S. aureus* (129B Strain)	*E. coli* (MG1655 Strain)	*P. aeruginosa* (2943 Strain)
R23I^T^	↓ (1 mg/mL)	↑	↑	↓ (10 mg/mL)
R23L^P^	↓ (0.1 mg/mL)	↓ (1 mg/mL)	↓ (0.1 mg/mL)	↓ (10 mg/mL)
V31K^T^	↓ (1 mg/mL)	−	−	−
R44K^S^	↓ (0.1 mg/mL)	↓ (10 mg/mL)	↓ (0.1 mg/mL)	↓ (10 mg/mL)
R44K^P^	↓ (0.1 mg/mL)	↑	↓ (0.1 mg/mL)	↓ (10 mg/mL)
V31K^S^	↓ (0.1 mg/mL)	↑	↓ (0.1 mg/mL)	↓ (10 mg/mL)
I31K^P^	↓ (1 mg/mL)	↓ (0.1 mg/mL)	−	↑

Notes: 
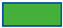
 ↓—Decrease in the parameter (optical density) compared to the negative control. 
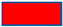
 −—No decrease in the parameter. 

 ↑—Reverse effect (increase in the optical density compared to the control).

**Table 2 ijms-26-08767-t002:** Determination of antibacterial properties of peptides on agar.

Peptide	Concentration	MRSA (SA 180-F Strain)	*S. aureus* (129B Strain)	*E. coli* (MG1655 Strain)	*P. aeruginosa* (2943 Strain)
R23I^T^	1 mg/mL	+	+	−	+
	0.1 mg/mL	−	−	−	−
	0.01 mg/mL	−	−	−	−
	0.001 mg/mL	−	−	−	−
R23L^P^	1 mg/mL	+	+	+	+
	0.1 mg/mL	−	−	−	−
	0.01 mg/mL	−	−	−	−
	0.001 mg/mL	−	−	−	−
V31K^T^	1 mg/mL	−	−	−	−
	0.1 mg/mL	−	−	−	−
	0.01 mg/mL	−	−	−	−
	0.001 mg/mL	−	−	−	−
R44K^S^	1 mg/mL	+	+	−	+
	0.1 mg/mL	−	−	−	−
	0.01 mg/mL	−	−	−	−
	0.001 mg/mL	−	−	−	−
R44K^P^	1 mg/mL	−	+	−	−
	0.1 mg/mL	−	−	−	−
	0.01 mg/mL	−	−	−	−
	0.001 mg/mL	−	−	−	−
V31K^S^	1 mg/mL	−	−	+	−
	0.1 mg/mL	−	−	−	−
	0.01 mg/mL	−	−	−	−
	0.001 mg/mL	−	−	−	−

Notes: “+”—The presence of a bacterial growth inhibition zone. “−”—The absence of a bacterial growth inhibition zone.

**Table 3 ijms-26-08767-t003:** Effect of peptides on early biofilm formation assessed by crystal violet staining and MTT assay.

Peptide	MRSA (SA 180-F Strain)	*S. aureus* (129B Strain)	*E. coli* (MG1655 Strain)	*P. aeruginosa* (2943 Strain)
**After c** **rystal v** **iolet s** **taining**				
R23I^T^	−	↑	↑	−
R23L^P^	↓ (10 mg/mL)	↑	↑	↓ (10 mg/mL)
V31K^T^	↑	↑	↑	−
R44K^S^	↓ (1 mg/mL)	↑	↑	↓ (10 mg/mL)
R44K^P^	↓ (10 mg/mL)	↑	↑	↓ (10 mg/mL)
V31K^S^	↓ (10 mg/mL)	↑	↑	↓ (10 mg/mL)
I31K^P^	−	↑	↑	−
**After MTT assay**				
R23I^T^	↑	↑	↑	−
R23L^P^	↑	↑	↑	↑
V31K^T^	↑	↑	−	−
R44K^S^	↑	↑	−	−
R44K^P^	↑	↑	↑	↓ (10 mg/mL)
V31K^S^	↑	↑	−	↓ (10 mg/mL)
I31K^P^	↑	↑	↑	−

Notes: 
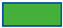
 ↓—Decrease in the parameter (optical density) compared to the negative control. 
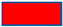
 −—No decrease in the optical density. 

 ↑—Reverse effect (increase in the optical density compared to the control).

**Table 4 ijms-26-08767-t004:** Effect of peptides on mature biofilm formation assessed by crystal violet staining and MTT assay.

Peptide	MRSA (SA 180-F Strain)	*S. aureus* (129B Strain)	*E. coli* (MG1655 Strain)	*P. aeruginosa* (2943 Strain)
**After c** **rystal v** **iolet s** **taining**				
R23I^T^	−	−	−	−
R23L^P^	−	↑	↑	↑
V31K^T^	−	↑	↑	↑
R44K^S^	↑	↑	↑	↑
R44K^P^	↑	↑	−	−
V31K^S^	↑	↑	−	↑
I31K^P^	↑	↑	−	↑
**After MTT assay**				
R23I^T^	−	−	↑	−
R23L^P^	↓ (0.01 mg/mL)	−	−	↑
V31K^T^	↓ (0.01 mg/mL)	−	−	↑
R44K^S^	↓ (0.01 mg/mL)	−	↑	↑
R44K^P^	↓ (0.01 mg/mL)	−	↑	−
V31K^S^	↓ (0.01 mg/mL)	−	↑	−
I31K^P^	↓ (0.01 mg/mL)	↓ (0.01 mg/mL)	−	↑

Notes: 
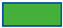
 ↓—Decrease in the parameter (optical density) compared to the negative control. 
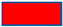
 −—No decrease in the optical density. 

 ↑—Reverse effect (increase in the optical density compared to the control).

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
