# Peer review of "In Vitro Evaluation of Antimicrobial Amyloidogenic Peptides for the Treatment of Early and Mature Bacterial Biofilms"

_ijms, 2025, doi:10.3390/ijms26188767_

Round 1

Reviewer 1 Report

Comments and Suggestions for Authors

The manuscript by Pavel A. Domnin and co-authors, titled "Evaluation of Antimicrobial Amyloidogenic Peptides for the Treatment of Early and Mature Bacterial Biofilms" is devoted to the investigation of the antimicrobial and antibiofilm activity of synthetic peptides derived from amyloidogenic sequences of the ribosomal protein S1. The authors evaluated the efficacy of seven peptides (R23IT, R23LP, V31KT, R44KS, R44KP, V31KS, I31KP) against biofilms formed by Gram-positive (Staphylococcus aureus, MRSA) and Gram-negative (Pseudomonas aeruginosa, Escherichia coli) bacteria. The study primarily focuses on the peptides' effects on early and mature stages of biofilm formation.  
The identified limitations (paradoxical effects, high working concentrations) highlight the need for further peptide optimization and mechanistic studies.  

The manuscript is well-written but there are minor remarks outlined below:  

- The study was conducted exclusively "in vitro". To confirm therapeutic potential, experiments on animal models are necessary. Therefore, it is recommended to include "in vitro" in the manuscript title and abstract to inform potential readers of the study's scope.  
- After the first mention, genus names should be abbreviated to the first letter (e.g., S. aureus, P. aeruginosa, E. coli): lines 58–59, 78, 112–114, 128, 334, and 338.
- Table 2 should be formatted to comply with the journal's guidelines.  
- In line 338, specify the resistance profile of strain 2943.  
- Include the manufacturers of key reagents (e.g., MTT, DMSO) in line 347.
- In line 385, provide the titles and a list of Supplementary Materials.

Reviewer 2 Report

Comments and Suggestions for Authors

This article investigates the antimicrobial and antibiofilm effects of several synthetic peptides derived with amyloidogenic propensity against clinically relevant bacteria, including MRSA, Pseudomonas aeruginosa, and E. coli. Among the tested peptides, R23LP and R44KS displayed strong and wide antimicrobial activity in both liquid and solid media. Biofilm assays revealed that peptide R44KS effectively inhibited early MRSA biofilm formation, while some peptides unexpectedly promoted biofilm growth in S. aureus and E. coli. For P. aeruginosa, peptides R44KP and V31KS reduced both biomass and metabolic activity in early-stage biofilms, and I31KP showed notable effects on reducing metabolic activity within mature MRSA biofilms. Overall, the study highlights stage- and strain-specific responses to these synthetic peptides and identifies several promising candidates for targeted biofilm control, particularly against MRSA. Very positive that not only the results compliant with the initial author's assumptions were mentioned and discussed.

Regarding the methodological approach of the authors, I believe several points should be clarified and provided in the text with more details.

Particularly, authors should provide answers to the following questions:

Were the peptides purified to a specific level (e.g., >95% HPLC purity)? If yes, how was purity verified?

How were peptides stored (solvent, temperature, lyophilized vs. dissolved)? Could storage conditions affect activity?

Were peptides used as-is from the supplier, or re-characterized (mass spec, analytical HPLC) to confirm identity before experiments?

Did the authors account for peptide solubility limits in aqueous medium (aggregation can be an issue at higher concentrations)?

What is the rationale for choosing these particular clinical isolates? Do they represent a broad enough panel of resistance phenotypes?

Were the resistance profiles confirmed independently by the authors or based solely on previous reports/clinical records?

Did they use reference ATCC strains alongside clinical isolates as controls?

Why were only 24 h biofilms tested? Biofilm maturation kinetics differ between species — did the authors optimize incubation times for each strain?

Did they normalize peptide concentrations to MIC values, or were concentrations chosen arbitrarily?

Crystal violet staining measures biomass (cells + matrix), while MTT measures metabolic activity — how did they control for cases where biofilm matrix remains but cells are dead (or vice versa)?

Was meropenem concentration chosen based on clinical breakpoint values or on preliminary MIC tests?

Did the authors quantify the baseline biomass of biofilms before treatment? Without this, it’s hard to know the relative reduction.

Were biofilms confirmed as mature (e.g., microscopy, thickness measurement), or is 24 h assumed sufficient?

Since biofilm eradication often requires higher concentrations than planktonic killing, were dose–response curves performed?

Why was 0.7% agar chosen (instead of the standard 1.5–2%)? Lower agar content can increase diffusion and may exaggerate inhibition zones.

How many replicates were performed per condition?

Were peptide diffusion properties considered? Some peptides may not diffuse well in agar, which could bias results.

What medium was used? Nutrient composition can affect peptide activity (e.g., cation concentration interferes with cationic peptides).

How were optical density measurements normalized to account for possible peptide absorbance or precipitation? (They mention baseline subtraction, but does this cover turbidity effects if peptides aggregate?)

Were bactericidal vs. bacteriostatic effects distinguished (e.g., via viable CFU counts, not just OD)?

I would also suggest clarifying the following potentially controversial or weakly substantiated statements in your Discussion:

How do you distinguish true peptide antibiofilm effects from artifacts such as peptide incorporation into the biofilm matrix that may bias crystal violet results?

What mechanistic explanation do you propose for peptides enhancing biofilm formation in S. aureus and E. coli? Could quorum-sensing mimicry be experimentally validated?

The strong differences between MRSA, P. aeruginosa, and E. coli suggest strain-specific factors. How generalizable are your findings across additional clinical isolates?

Could the observed increases in metabolic activity in some biofilms be due to hormesis (stimulation at low doses)? How might this affect therapeutic applications?

Since activity was mainly observed at 1 mg/mL, how physiologically relevant are these concentrations for in vivo applications?

Do you have evidence that the peptides act via membrane disruption, or might they also modulate bacterial signaling pathways?

You mention structural optimization and synergistic combinations. Which chemical modifications or combination strategies do you consider most promising?
